# One-step generation of multiple gene knock-outs in the diatom *Phaeodactylum tricornutum* by DNA-free genome editing

Manuel Serif[1,2,3], Gwendoline Dubois[1,2,3], Anne-Laure Finoux[1,2,3], Marie-Ange Teste[1,2,3], Denis Jallet[1,2,3] & Fayza Daboussi[1,2,3]

Recently developed transgenic techniques to explore and exploit the metabolic potential of microalgae present several drawbacks associated with the delivery of exogenous DNA into the cells and its subsequent integration at random sites within the genome. Here, we report a highly efficient multiplex genome-editing method in the diatom *Phaeodactylum tricornutum*, relying on the biolistic delivery of CRISPR-Cas9 ribonucleoproteins coupled with the identification of two endogenous counter-selectable markers, *PtUMPS* and *PtAPT*. First, we demonstrate the functionality of RNP delivery by positively selecting the disruption of each of these genes. Then, we illustrate the potential of the approach for multiplexing by generating double-gene knock-out strains, with 65% to 100% efficiency, using RNPs targeting one of these markers and *PtAureo1a*, a photoreceptor-encoding gene. Finally, we created triple knock-out strains in one step by delivering six RNP complexes into *Phaeodactylum* cells. This approach could readily be applied to other hard-to-transfect organisms of biotechnological interest.

[1] INSA, UPS, INP, LISBP, Université de Toulouse, 135 Avenue de Rangueil, F-31077 Toulouse, France. [2] INRA, LISBP, UMR792, 135 Avenue de Rangueil, F-31077 Toulouse, France. [3] CNRS, LISBP, UMR5504, 135 Avenue de Rangueil, F-31077 Toulouse, France. These authors contributed equally: Manuel Serif, Gwendoline Dubois. Correspondence and requests for materials should be addressed to F.D. (email: fayza.daboussi@insa-toulouse.fr)

The term microalgae encompasses a vast diversity of organisms, which make major contributions to biogeochemical cycles on a global scale[1]. A very attractive aspect concerns the ability of most microalgae to perform photosynthesis, allowing the generation of biological compounds from carbon dioxide ($CO_2$) as an inorganic carbon source and light as an energy source. Thus, these versatile cell factories have the potential to become key enablers for industrial biotechnology. Today, only a few microalgae species are commercially exploited[2]. They include the diatom *Phaeodactylum tricornutum*, which naturally synthesizes numerous marketable compounds[3], such as pigments and long-chain omega-3 fatty acids[4,5].

Genetic engineering is a promising approach to improve the metabolic potential of microalgae and obtain further insights into their metabolism and physiology. Custom molecular scissors have recently emerged as useful tools to induce site-specific modifications to the genome of diatoms[6–12]. These nucleases recognize and introduce a double strand break (DSB) at a target genomic sequence, which is repaired by non-homologous end joining (NHEJ) with potentially associated targeted mutagenesis (TM) or by homology-directed repair (HDR) in the presence of a DNA donor template[13]. To date, meganucleases, TALE nucleases, and the CRISPR-Cas9 system have all been successfully applied to inactivate single target genes in *Phaeodactylum*, either for functional analysis[6–12] or to redirect natural metabolism towards increased neutral lipid biosynthesis[6]. However, the construction of complex synthetic metabolic pathways requires the simultaneous introduction of multiple genetic modifications, as exemplified by the ten genome changes required to generate a hydrocortisone-producing yeast[14]. Such a challenge has not yet been addressed by the microalgal community.

Indeed, the "nuclease-driven" genetic engineering of diatoms is still in its infancy. Until now, it has been mediated by transforming plasmids encoding a nuclease and an antibiotic resistance cassette into the cells, both then stably integrated at random sites within the nuclear genome[6–11]. Disadvantages of this approach include: the low transformation efficiencies (less than $10^{-6}$); the long-term expression of the nuclease can potentially induce off-target cleavage[15;] the random integration of all or part of the plasmid DNA into the genome can lead to undesired gene disruptions or uncontrolled effects on gene expression near the integration site(s); and the impossibility to eliminate background mutations or integrated transgenes through outcrossing in *Phaeodactylum*, as it is a diploid organism with no known sexual reproduction. Alternative genome editing methodologies must be developed to circumvent these issues. Recently, an episome-based protocol that avoids genomic integration has been described[12] that still elicits Cas9 expression over several generations, with potential toxicity or off-target effects[15]. Another possibility is to establish DNA-free genome-editing approaches, as illustrated in only one microalga thus far: the chlorophyte *Chlamydomonas reinhardtii*[16–19]. In *Chlamydomonas*, the direct transformation of Cas9/ single-guide RNA (sgRNA) ribonucleoprotein complexes (RNPs) can drive TM, but requires the use of cell-wall less mutants[18,19] or a plasmid carrying an antibiotic resistance cassette must be co-transformed with the RNP for selection[17]. The challenge in *Phaeodactylum* rests on both the introduction of RNP complexes through the complex cell wall and the development of an antibiotic-free selection method to enrich for transformants harboring TM events. We thus propose a strategy relying on the simultaneous co-delivery of multiple RNP complexes by biolistic, one targeting an endogenous gene for which inactivation confers positive selection, and the others targeting genes of interest.

Here, we identify two endogenous marker genes in *Phaeodactylum*, one homologous to *URA3*[20], the other to *APT*[21]. We demonstrate that their inactivation, by either nucleases expressed from the genome or RNP complexes, leads to 5-fluoroorotic acid (5-FOA) and 2-fluoroadenine (2-FA) resistance, a property that can be used for the direct selection of the transformants. Combining RNPs targeting either of these marker genes and another gene of interest, *PtAureo1a*, encoding a blue-light photoreceptor/ transcription factor[22], we generated multiple knock-outs in a single step at high frequency. Our results illustrate that "DNA-free" genome-editing can be achieved in diatoms. We anticipate that it can be transferred to any other hard-to-transfect organism, as it relies on well conserved endogenous selection markers.

## Results

**Inactivation of the *UMP synthase* gene using TALENs.** We first evaluated the use of endogenous positive selectable markers for DNA-free genome editing by attempting to produce knock-out (KO) strains of the *PtUMPS* gene (Phatr3_J11740), the *Phaeodactylum URA3* homolog of which knock-down increases 5-FOA tolerance while leading to uracil auxotrophy[20]. We designed a TALEN pair to target the orotidine-5′-phosphate decarboxylase domain-encoding sequence within *PtUMPS* exon 1 (Fig. 1a). The corresponding TALEN-encoding plasmids were assembled and co-delivered by biolistic bombardment into wild-type (WT) *Phaeodactylum* cells (NCMA strain), together with a plasmid conferring resistance to nourseothricin (NAT). Of the 30 NAT-resistant colonies that appeared, five showed successful genomic integration of the TALEN monomer genes, as revealed by PCR. These colonies were then spotted onto 5-FOA selective medium supplemented with uracil[20]; two of five colonies, hereafter referred to as 8A2 and 12A1, grew in this condition. Genomic PCR amplification of *PtUMPS* from 8A2 and 12A1 revealed the presence of mixed cell populations, a phenomenon referred to as mosaicism[13] (Fig. 1b). Based on chromatogram deconvolutions, the TIDE calculation software[23] suggested a variety of INDELs at the *PtUMPS* locus for the 5-FOA resistant colonies (Fig. 1c). Therefore, we subcloned the 8A2 and 12A1 populations on medium containing 5-FOA and uracil and then sequenced the *PtUMPS* locus in 36 individual subclones for 8A2 and 20 individual subclones for 12A1. As expected, all showed mutagenic events (examples are depicted in Fig. 1d). In 19 of 20 12A1 subclones, both *PtUMPS* alleles showed mutagenic events based on polymorphism patterns present upstream and downstream of the target site (Supplementary Fig. 1). Only one allele was amplified in 35 of 36 8A2 subclones and 1 of 20 12A1 subclones (Fig. 1d), suggesting large insertions or deletions in the other allele[6,9], a phenomenon known as loss of heterozygosity (LOH)[24]. In the detected allele, INDELs were systematically present.

We further characterized two 8A2 subclones, 8A2_1 and 8A2_8, and two 12A1 subclones, 12A1_2 and 12A1_8, for their ability to grow in liquid F/2 medium with or without uracil and with or without 5-FOA (Fig. 1e–g; Supplementary Fig. 2). The WT strain (NCMA) and a colony (2A3) only transformed with the NAT[R] plasmid were used as controls. All subclones carrying mutations in *PtUMPS* exhibited growth in the presence of 5-FOA and uracil (Fig. 1e), in contrast to the two controls. This confirms the expected phenotype for the *PtUMPS* KO concerning 5-FOA resistance. The quasi-absence of growth observed for the *PtUMPS* mutants in the absence of uracil confirmed their auxotrophy for uracil (Fig. 1f). Importantly, their growth kinetics in medium supplemented with uracil was equivalent to that of controls cultivated in the presence or absence of uracil (Fig. 1g). This suggests that the *PtUMPS* mutation has no impact on cell fitness under our culture conditions, a major prerequisite for further laboratory and industrial applications. We thus confirmed that *PtUMPS* KO leads to 5-FOA[R] and uracil auxotrophy, a phenotype that can be used for direct selection.

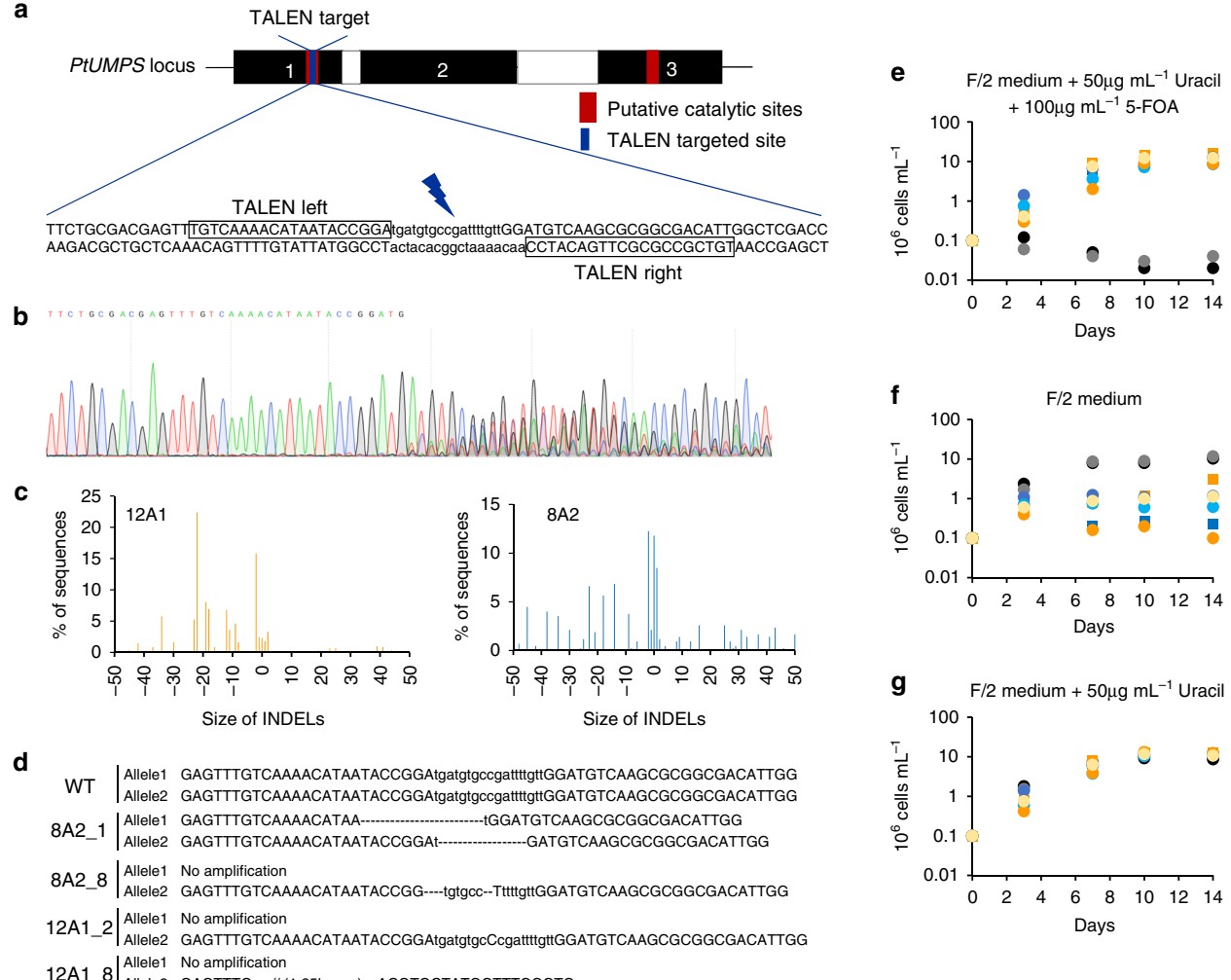

**Fig. 1** Characterization of *PtUMPS* knock-out strains generated using a TALE nuclease. **a** Structure of the *PtUMPS* locus. The exons are represented as black boxes, with the putative catalytic sites in red and the TALEN-targeted site in blue. The half-sites recognized by each TALEN are boxed. **b** Sequence chromatogram of a PCR product amplified from strain 12A1, showing the presence of heterogeneous sequences surrounding the TALEN cutting site. **c** INDEL spectrum and frequency predicted by the Track INDELs by DEcomposition (TIDE) algorithm in the two 5-FOA-resistant colonies, 8A2 and 12A1. **d** Example of mutagenic events through the alignment of the wild type and 8A2_1, 8A2_8, 12A1_2, and 12A1_8 subclones DNA sequences at the *PtUMPS* locus. **e–g** Representative growth experiments from two independent repeats performed over 14 days with the parental NCMA strain (black circles), the 2A3 control strain transformed with the NAT vector only (gray circles), two 5-FOA-resistant populations (8A2 dark blue squares and 12A1 dark orange squares), and four subclones: 8A2_1 (dark blue circles), 8A2_8 (light blue circles), 12A1_2 (dark orange circles) and 12A1_8 (yellow circles), **e** F/2 medium supplemented with 50 µg mL⁻¹ uracil and 100 µg mL⁻¹ 5-FOA, **f** classical F/2 medium, or **g** F/2 supplemented with 50 µg mL⁻¹ uracil. Y-axis **e–g** presents cell density, expressed as million cells per mL, on a logarithmic scale

**Antibiotic-free Cas9-driven inactivation of the *PtUMPS* gene.** Other prerequisites for our proposed DNA-free genome editing strategy consist of demonstrating that (i) a *PtUMPS* KO can be achieved efficiently without introducing an exogenous antibiotic resistance cassette, solely relying on the 5-FOA$^R$ phenotype for selection and (ii) multiple genes can be efficiently inactivated in a single step, including *PtUMPS*, which serves as an endogenous co-selection marker. The versatility of the CRISPR-Cas9 system, associated with its potential for multiplexing, prompted us to evaluate its capacity to achieve this strategy. We designed three gRNAs directed towards sequences situated within *PtUMPS* exons 1 and 3 (gUMPS1 and gUMPS4, Fig. 2a), which encode the putative catalytic sites, and exon 2 (gUMPS3) using the CRISPOR web-based tool[25].

We tested whether the selected gRNAs can efficiently target the Cas9 protein towards the *PtUMPS* locus by first performing in vitro assays. gRNAs were complexed to the recombinant Cas9

protein (Supplementary Fig. 3), thereby forming ribonucleoproteins (RNPs). The complexes were then mixed with a *PtUMPS* amplicon comprising the target sites. All RNP complexes efficiently catalyzed double strand DNA cleavage of the *PtUMPS* fragment, whereas Cas9 or gRNAs alone did not (Supplementary Fig. 3).

Three separate sgRNA expression vectors were assembled for gUMPS1, gUMPS3, and gUMPS4, as previously described[10], to examine the in vivo functionality of these gRNAs. Individual sgRNA expression plasmids were co-bombarded into WT cells together with a plasmid encoding the Cas9 nuclease and another carrying the NAT$^R$ cassette. NAT$^R$ colonies appeared under all transformation conditions after 2–3 weeks. The presence of the Cas9-encoding gene was verified in all transformants by PCR before the *PtUMPS* locus was sequenced. From 29 to 100% of the screened colonies showed TM at the *PtUMPS* locus, depending on the gRNA used (Fig. 2b). Examples of mutagenic events are

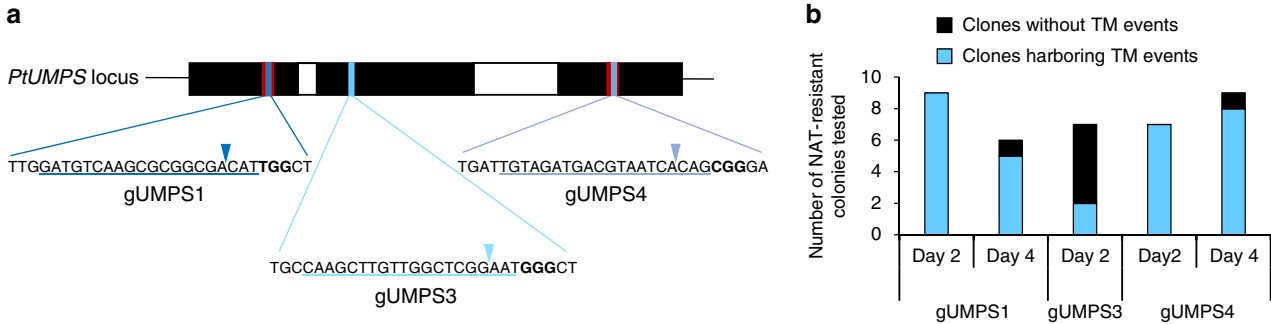

**Fig. 2** CRISPR/Cas9-induced knock-out mutations at the *PtUMPS* locus. **a** Schematic representation of the position and sequence targeted by each gRNA: gUMPS1, gUMPS3, and gUMPS4. The 20-nt region of the target site is underlined and the PAM (Protospacer Adjacent Motif) is shown in bold letters. **b** Efficiency of targeted mutagenesis induced by the delivery of the NAT resistance cassette, gRNA, and Cas9 encoding vectors into WT *Phaeodactylum* cells after selection on NAT containing medium

illustrated in Supplementary Figs. 4, 5 and 6. We only detected one allele for most clones, again indicative of LOH; INDELs appeared close to the PAM sequences on the other allele, as expected. Thus, this step validated gUMPS1, gUMPS3, and gUMPS4 function in vivo.

Next we examined whether it was possible to obtain 5-FOA-resistant colonies after direct selection of the cells transformed with plasmids encoding the Cas9 protein and either gUMPS1 or gUMPS4. Cells were transformed with either of these gRNA encoding plasmids and directly spread onto selective medium containing uracil and 5-FOA. Twelve 5-FOA$^R$ clones appeared. Their genotypic characterization confirmed the presence of mutations adjacent to the PAM sequence in 10 out of 12 cases (Supplementary Table 1), demonstrating that direct selection on 5-FOA can lead to the identification of mutagenic events. The two remaining clones were not mutated at the *PtUMPS* loci, suggesting that they were false positives.

**Demonstration of RNP delivery in *Phaeodactylumtricornutum*.** We established the proof of concept of DNA-free genome editing in diatoms by adapting a proteolistic protocol established by Martin-Ortigosa and Wang[26] to deliver Cas9–gRNA RNPs into *Phaeodactylum* cells (Fig. 3a). We first evaluated the frequency of targeted mutagenesis induced by the Cas9-gUMPS1 RNP complex in the absence of selection. To achieve that, *P. tricornutum* cells were bombarded with a dose response of 0, 2, 4, and 8 µg of Cas9-gUMPS1 complex. We did not expect a RNP transformation efficiency higher than 10$^{-6}$ in the absence of selection, as in the case of DNA delivery it is necessary to co-transform 100 million cells with a plasmid encoding an antibiotic resistance to obtain 30 antibiotic-resistant clones[6,9,10]. To test this hypothesis, we collected the cells growing in the absence of selection at four days post-bombardment and quantified the mutagenesis at the *PtUMPS* locus using locus-specific PCR followed by deep sequencing (Supplementary Fig. 7). As positive controls, different amounts of a monoallelic mutant strain carrying a 1nt deletion at the gUMPS1 target site were mixed with WT cells to get cell-to-cell ratios of 100%, 10%, 1%, 0.1% and 0%. Whereas mutagenic events were detected in the positive controls at the expected frequencies, no induced mutagenic event was detected in the samples corresponding to cells bombarded with Cas9-gUMPS1, which reveals a TM frequency of less than 0.4%. This result suggests that in the absence of selection, it is quasi impossible to produce DNA-free transgenic strains. We decided to evaluate our ability to generate such mutant strains using the *PtUMPS* counter-selectable marker.

The gUMPS1, gUMPS3, and gUMPS4 guides were individually mixed with the Cas9 protein to form the RNP complexes. Four

micrograms of individual Cas9–gRNA RNP complex was coated onto gold particles and bombarded into WT *Phaeodactylum* cells using a helium gene gun. Cultures were plated onto 5-FOA plus uracil containing medium two or four days post-bombardment. After three to four weeks, 5-FOA resistant colonies appeared for all conditions, except when cells were bombarded with Cas9 only as a negative control (Supplementary Table 2). A second round of selection confirmed that four of four colonies obtained with the gUMPS4 RNP complex, one of three with the gUMPS1 RNP complex, and one of two with the gUMPS3 RNP complex could grow on 5-FOA uracil medium. These 5-FOA$^R$ strains were then analyzed for TM at the *PtUMPS* locus (Supplemental Table 1). We failed to amplify the *PtUMPS* locus in the gUMPS1-derived and gUMPS3-derived strains, suggesting that a large deletion may have occurred. However, we observed TM close to the PAM in all gUMPS4-derived strains. This shows that transient exposure to the RNPs is sufficient to induce TM at an endogenous locus. Thus, we generated knock-out transgenic strain without introducing exogenous DNA into the cells.

**Optimizing multiplex DNA-free genome editing.** Our main objective was to inactivate multiple genes of interest in a single step, relying on the *PtUMPS* KO 5-FOA-resistance phenotype for positive selection. We started with the assumption that double KO strains could be generated by bombarding cells with a combination of RNP complexes, targeting the *PtUMPS* gene and another gene of interest (Fig. 3a). We selected the gene encoding Aureochrome 1a as an example (*PtAureo1a*, *Phatr3_J8113*); PtAureo1a is a photoreceptor that participates in blue light perception[22] and whose TALEN-mediated biallelic genetic KO was previously reported[9,11]. The gRNAs targeting *PtAureo1a* (gAureo1a2 and gAureo1a3) were selected following the same in silico criteria as for *PtUMPS* (Supplementary Table 3). Their activities were first verified in vitro using the RNP-based assay (Supplementary Fig. 8). Next, WT *Phaeodactylum* cells were simultaneously bombarded with multiple RNP complexes, targeting both *PtUMPS* and *PtAureo1a* (Fig. 3a).We used two gRNAs per target (gUMPS1 and gUMPS3, targeting the *PtUMPS* locus; gAureo1a2 and gAureo1a3, targeting the *PtAureo1a* locus) and mixed them independently with recombinant Cas9 protein to increase our chances of obtaining TM at both loci. The four RNP complexes were then combined (Fig. 3a). Cells were then co-bombarded with 8 µg of RNP complexes (corresponding to 2 µg of each RNP) and transferred onto 5-FOA uracil medium two days post-bombardment. Clones were first analyzed for TM at the *PtUMPS* locus (Fig. 3b). Among the 19 5-FOA$^R$ colonies that appeared, 17 (89%) showed TM at the *PtUMPS* locus: 4/17 (23%) with mutagenic events on both alleles and 13/17 (76%) with mutagenic

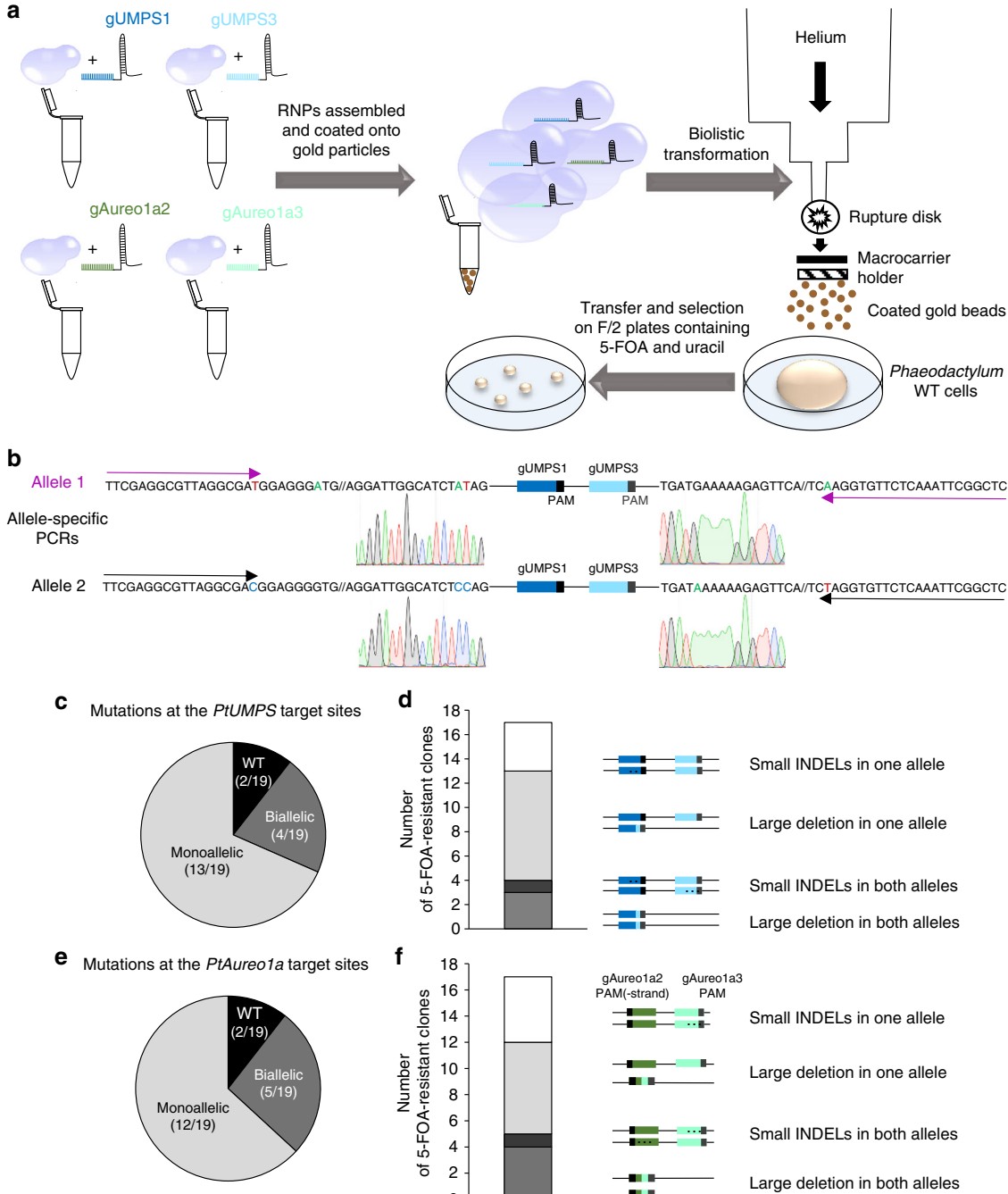

**Fig. 3** Multiple gene knock-outs using a DNA-free RNP genome editing approach. **a** Outline of the genome-editing workflow used here. The first step consists of generating four independent RNP complexes, two of them targeting *PtUMPS* and the two others targeting *PtAureo1a* or any other gene of interest. Each RNP complex is prepared in a separate tube by incubating the Cas9 protein with pre-annealed crRNA::tracrRNA complexes (crRNA: base pairing site, tracrRNA: trans-activating crRNA). Next, the four RNP complexes are combined and loaded onto gold particles. The third step consists of the delivery of these coated gold beads into *Phaeodactylum* cells using a biolistic apparatus. Finally, after two to four days, the transformed cells are re-plated onto selective medium, here F/2 medium supplemented with uracil and 5-FOA. After three to four weeks, 5-FOA-resistants colonies appear and are analyzed for the presence of mutagenic events. **b** Each allele of the *PtUMPS* gene was amplified by allele-specific PCRs from the obtained 5-FOA resistant colonies and sequenced. **c** Frequency of colonies harboring no mutagenic event at the *PtUMPS* loci (black sector on the pie chart) or a mutagenic event on one allele (light gray sector) or both alleles (dark gray sector). **d** Additional information on the nature of the mutagenic events is reported, with four classes of mutagenic events being illustrated: small INDELs in one allele, the second being WT (white bar); large deletion corresponding to the DNA sequence between the two targeted sites in one allele, the second allele being WT (light gray bar); small INDELs in both alleles (dark gray bar); deletion of the DNA fragment between the two targeted sites in both alleles (medium gray bar). **e** Frequency of targeted mutagenesis of the *PtAureo1a* target sites (black sector: no TM detected, light gray sector: monoallelic mutants, dark gray sector: biallelic mutants). All clones mutated in *PtUMPS* also carried a mutation in at least one of the *PtAureo1a* alleles. **f** Distribution for the various classes of mutagenic events at the *PtAureo1a* loci

events on one allele only, the second carrying the WT sequence (Fig. 3c). Amplification of *PtUMPS* resulted in a clearly shorter PCR product for 11/17 (65%) of the clones (Fig. 3d, Supplementary Fig. 9). Sequencing of these PCR products revealed deletion of the fragment between gUMPS1 and gUMPS3, indicating that DSBs were created at both target sites (Fig. 3d, Supplementary Fig. 9). The *PtAureo1a* locus of the 19 5-FOA[R] colonies was amplified by PCR and sequenced. The two strains that did not show TM of *PtUMPS* also showed no TM of *PtAureo1a*. Further analysis revealed that 100% of the *PtUMPS*-mutated clones also displayed a mutagenic event at the *PtAureo1a* locus; 5 of 17 carried mutations on both alleles (Fig. 3d, e). Thus we achieved simultaneous multiple gene knock-outs in diatoms.

**Extrapolation to another selectable marker**. We evaluated the transferability of this approach by investigating another putative selectable marker, the *Adenine Phosphoribosyl Transferase (APT)* gene, for which inactivation confers 2-FA resistance in several organisms, including plants[21,27]. APT normally contributes to adenine recycling as part of the nucleotide salvage pathway[27]. We identified a single potential APT encoding gene in the *Phaeodactylum* genome (*Phatr3_J6834*, hereafter referred to as *PtAPT*; UniProt B7G7K1), based on sequence homology of its translation product with the *Physcomitrella patens* APT protein (PpAPT, UniProt: Q45RT2)[21]. Overall, PtAPT shares 41.7% amino-acid identity with PpAPT and 28.6% with the *Arabidopsis thaliana* APT1 protein (AtAPT1, UniProt: P31166) (Fig. 4a). The phosphoribosyl transferase (PRT) type I domain, spanning amino-acid positions 28 to 132 in PtAPT, is well conserved.

Before proceeding further, we determined whether 2-FA was toxic to *Phaeodactylum* cells and at what concentrations. We observed a dose-response effect of 2-FA on growth, with complete inhibition at 10 μM (Fig. 4b).

We designed gRNAs directed against *PtAPT* (gAPT1 and gAPT3) to target regions within exons 1 and 2, respectively (Fig. 4c). Both the gAPT1 RNP and gAPT3 RNP complexes efficiently catalyzed DSB DNA cleavage of a *PtAPT* amplicon in vitro (Supplementary Fig. 10). Either the gAPT1 or the gAPT3 RNP complex was then introduced into WT *Phaeodactylum* cells by proteolistic transformation. Cultures were plated onto selective medium containing 10 μM 2-FA and 5 mg.L$^{-1}$ adenine, two days post-bombardment (Fig. 4d). Eleven and five 2-FA[R] colonies appeared after two weeks for the gAPT1 and gAPT3 RNPs, respectively, whereas no colonies appeared in the negative control bombarded with the Cas9 protein alone. Among the 16 2-FA[R] colonies, 15 (93%) showed TM at the *PtAPT* locus (Fig. 4d), all of which carried mutations on both alleles, suggesting that biallelic KO is required to trigger resistance to 2-FA. Examples of mutagenic events are shown in the Supplementary Figs. 11 and 12. We thus demonstrated that inactivation of *PtAPT* confers 2-FA[R], allowing the use of *PtAPT* as an endogenous positive selectable marker in *Phaeodactylum*.

We next evaluated the impact of knocking out *PtAPT* on growth in F/2 medium, F/2 medium supplemented with adenine, and F/2 medium supplemented with both 2-FA and adenine (Fig. 4e). As expected, WT cells were unable to grow in the presence of 10 μM 2-FA. The two tested APT KO strains were unaffected by the addition of 2-FA and grew normally in F/2 or F/2 plus adenine. Inactivating *PtAPT* did not negatively affect culture fitness, an important requirement for downstream applications.

We verified that the *PtAPT* KO can be used as a co-selection marker for multiplexing experiments by simultaneously bombarding WT *Phaeodactylum* cells with RNP complexes targeting the *PtAPT* and *PtAureo1a* loci (Fig. 4f). Two gRNAs were used per target (gAPT1 and gAPT3 targeting the *PtAPT* locus;

gAureo1a2 and gAureo1a3 targeting the *PtAureo1a* locus). Thus, cells were bombarded with 8 μg of an RNP mixture consisting of all four complexes and spread onto selective medium containing 2-FA and adenine two days post-bombardment. Thirty-four 2-FA[R] clones appeared within two weeks. All showed TM at the *PtAPT* locus. Amplification of *PtAPT* resulted in shortened PCR fragments for 47% of the colonies (16/34), indicating that the deletion occurred between the gAPT1 and gAPT3 target sites (Supplementary Fig. 13). These results were confirmed by sequencing 29 of these 34 clones. We also investigated the presence of mutagenic events at the *PtAureo1a* locus. Sixty-five percent (19/29) of the colonies harboring *PtAPT* mutations also showed mutagenic events at the *PtAureo1a* locus and 52% (10/19) of them exhibited a deletion of the fragment between the two target sites. Interestingly, 52% (10/19) of the *PtAureo1a* mutants were mutated on both alleles, leading to the generation of PtAureo1a KO strains (Fig. 4f).

We next determined whether three genes can be simultaneously knocked-out. Cells were bombarded with a mixture of six RNP complexes, two RNPs targeting *PtAPT*, two *PtUMPS*, and two *PtAureo1a*, and selected on 2-FA medium supplemented with adenine and uracil. Fourteen 2-FA-resistant colonies appeared. Thirteen of the fourteen colonies (93%) showed a mutagenic event at the *PtAPT* locus (Fig. 5). Among the 13 clones, five were mutated at both *PtAPT* and *PtAureo1a*, two at both *PtAPT* and *PtUMPS*, and two at all the three loci. The genotype of these clones is summarized in Fig. 5.

We should point out that the selection of 2-FA resistant clones was achieved in all of our independent trials ($n = 7$) whereas the selection of 5-FOA-resistant clones failed in three experiments out of eight. Such a disparity has already been reported in other organisms where pH, light, and temperature have been shown to influence 5-FOA selection strength. The major point is that we have succeeded in generating multiple *PtAureo1a* knock-out strains in the three independent multiplexed experiments described in Supplementary Table 4. Our results show that this genome-editing approach is a powerful tool to simultaneously inactivate multiple genes without the use of DNA and antibiotics.

To go one step further, we phenotypically characterized eleven RNP-derived transformants from several independent experiments that were mutated in *PtUMPS* and/or *PtAPT* and/or *PtAureo1a* (Supplementary Fig. 14a). First we tested their resistance to 5-FOA, as well as 2-FA (Supplementary Fig. 14b). For all these strains, sensitivity or resistance to different media reflected genotypes well. Next, we quantified the amount of PtAureo1a in the strains by Western Blotting (Supplementary Fig. 15). Whereas we did not detect any PtAureo1a protein in the samples derived from bi-allelic *PtAureo1a* mutant strains, we clearly detected it in protein samples derived from the tested monoallelic PtAureo1a strain and the positive controls. We additionally performed growth experiments and observed that the cells mutated in *PtAureo1a* had lower (20% maximum) growth rates compared to the parental strain or the *PtUMPS* and *PtAPT* single mutants (Supplementary Fig. 16). Finally, we evaluated the photo-physiological impact of the PtAureo1a mutation. This gene encodes for a blue-light photoreceptor of which knock-out strongly affects cells ability to dissipate excess light energy as heat upon a shift from low to high irradiances[9]. This can be quantified by measuring the Non-Photochemical Quenching (NPQ) of chlorophyll fluorescence[9]. As previously reported, biallelic *P. tricornutum PtAureo1a* mutant strains generated by TALE nucleases display reduced NPQ capacities compared to wild-type[9]. We performed similar NPQ capacity measurements. Whereas no NPQ phenotype was observed in the monoallelic *PtAureo1a* mutant strain tested here and in the PtAPT and PtUMPS knock-out strains, we measured a 50–73% reduction of

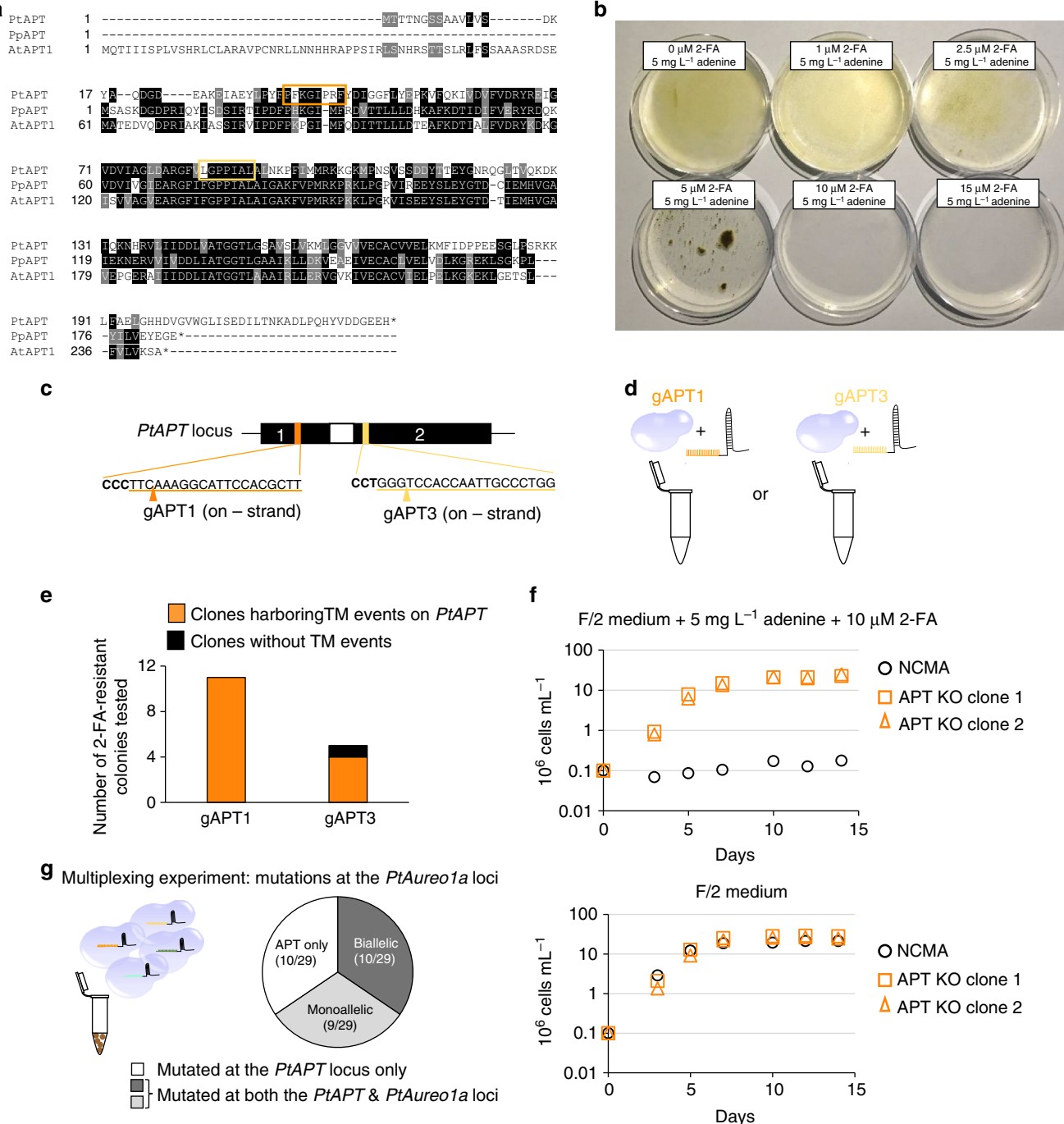

**Fig. 4** *PtAPT*, another endogenous selection marker for DNA-free genome editing. **a** Alignment of the APT protein sequences from *Arabidopsis thaliana* (AtAPT1), *Physcomitrella patens* (PpAPT), and *P. tricornutum* (PtAPT). Conserved amino acids are highlighted. Orange boxes represent the gRNA targeting sites. **b** Sensitivity of NCMA (WT) *P. tricornutum* cells to chronic exposure to various concentrations of 2-FA in the presence of adenine. **c** Structure of the *PtAPT* locus. Exons are represented as black boxes, introns as white boxes. Sequences targeted by gAPT1 (dark orange) and gAPT3 (light orange), both of them on the reverse complementary (−) strand, are indicated with their Protospacer Adjacent Motives shown in bold font. **d** RNPs were prepared by complexing either of the corresponding gRNAs to recombinant Cas9. **e** In vivo evaluation of targeted mutagenesis induced by the Cas9–gAPT1 or Cas9–gAPT3 RNP complexes. *Phaeodactylum* cells bombarded with the corresponding RNP were selected on F/2 medium containing 10 μM 2-FA plus 5 mg L$^{-1}$ adenine. The *PtAPT* locus of the generated 2-FA-resistant colonies was amplified by PCR and sequenced. **f** Representative growth experiment from two independent repeats performed with two bi-allelic *APT* mutants and the NCMA parental cell grown in F/2 plus 2-FA and F/2 alone. **g** Frequency of targeted mutagenesis of the *PtAureo1a* loci in colonies that were simultaneously bombarded with two RNPs against *PtAPT* (gAPT1 and gAPT3) and two RNPs against *PtAureo1a* (gAureo1a2 and gAureo1a3)

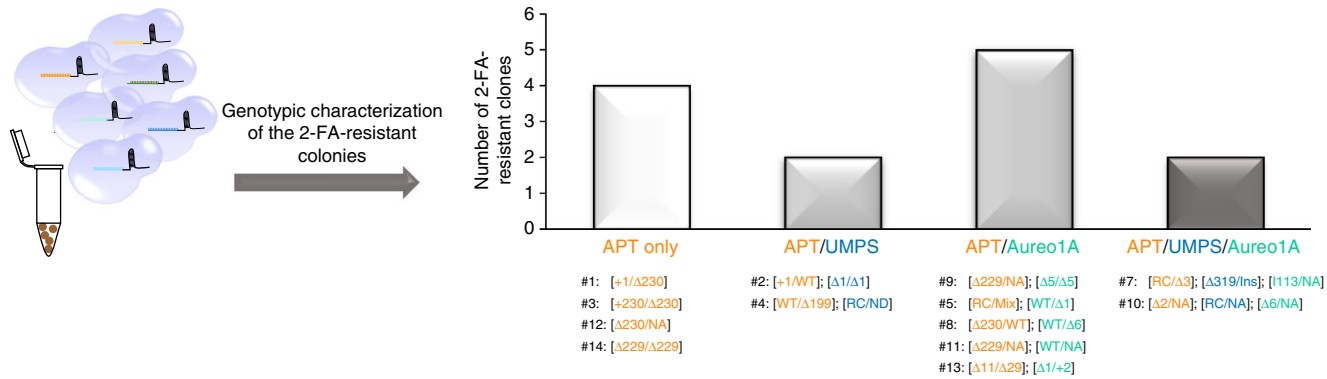

**Fig. 5** One-step generation of triple knock-out strains. WT *Phaeodactylum* cells were simultaneously bombarded with six RNP complexes, each directed against one specific target at either the *PtAPT*, *PtUMPS*, or *PtAureo1a* loci. After two days, cultures were transferred onto 2-FA selective medium supplemented with adenine and uracil. Fourteen 2-FA resistant clones (referred to as strains AUA#1 to #14) were obtained and screened for the presence of mutagenic events at the six target sites. Bar plots present the number of 2-FA resistant colonies harboring mutations at the *PtAPT* loci only (white bar), at both the *PtAPT* and *PtUMPS* loci or the *PtAPT* and *PtAureo1a* loci (light gray bars), or at all three loci (dark gray bar). The genotypes of the corresponding clones are presented underneath the graph and written in orange font for *PtAPT*, blue for *PtUMPS* and green for *PtAureo1a*. RC reverse complement insertion. NA not amplifiable, Mix mosaic, ND not determined

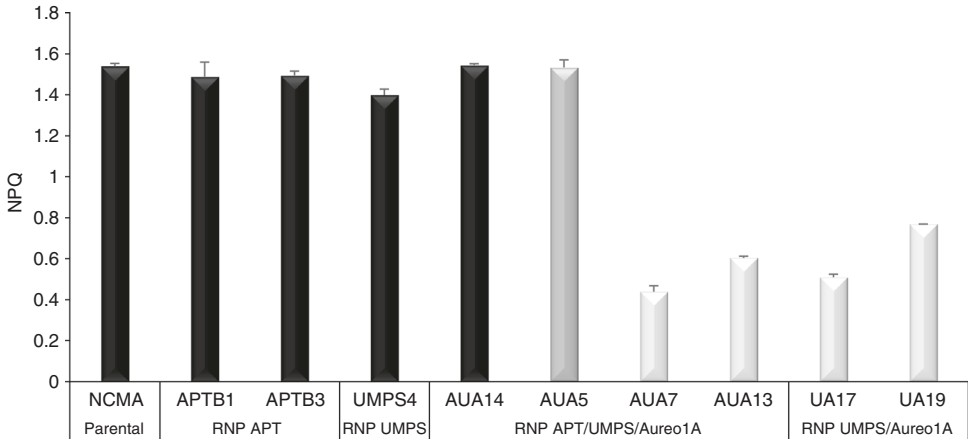

**Fig. 6** Non-photochemical quenching in RNP-derived *PtAureo1a* mutant strains. The non-photochemical quenching (NPQ) of chlorophyll fluorescence was measured after ten minutes of illumination with 700 μmol photons m$^{-2}$ s$^{-1}$ of actinic red light. The parental NCMA strain, two *PtAPT* single mutants (APTB1 and APTB3), a *PtUMPS* single mutant (UMPS4) and strain AUA14 showing no TM at the *PtAureo1a* loci served as controls (black bars). Several strains derived from multiplexing experiments—including a *PtAureo1a* monoallelic mutant (AUA5; dark gray bar), as well as *PtAureo1a* biallelic mutants (AUA7, AUA13, UA17, and UA19; light gray bars)—were characterized. The genotypes of these strains are shown in Supplementary Fig. 14. Data are the average of $n = 2$ independent repeats, bars show SD

NPQ in all RNP-derived biallelic *PtAureo1a* mutants tested (Fig. 6). Therefore, we confirmed the loss of function of the generated PtAureo1a knock-out strains, which reveals the power of our approach to study gene function.

## Discussion

Here, we established a DNA-free approach that solves the important issues of random DNA integration into the diatom genome and long-term nuclease expression. This was achieved by combining RNPs targeting one endogenous marker gene with RNPs targeting another gene of interest, successfully generating dozens of transgenic strains free of foreign DNA, without the use of antibiotics. Moreover, the same strategy allowed the creation of triple-gene knock-out strains in a one-step procedure.

This is a valuable research tool for algae genome engineering that should open the way to the study of putative redundant functions of multiple gene family members. For example, it could be suitable to investigate the role of specific light-harvesting complexes of the LHCX family during exposure to excess

light[28] or the contribution of single aureochromes to photoacclimation[11].

Our approach also represents a technical breakthrough for industrial biotechnology, as it will permit the simultaneous KO of multiple endogenous pathways to redirect metabolism towards the production of biomolecules of interest. Moreover, the success of our strategy was associated with the identification of the *PtUMPS* and *PtAPT* genes and the validation that their inactivation confers resistance to specific molecules. The use of such markers represents an attractive alternative to conventional antibiotic-resistance genes and solves the environmental issue posed by the dissemination of such resistance genes into other organisms through horizontal transfer.

The methodology we have developed addresses the global effort to exploit non-integrative expression of the CRISPR-Cas9 system. A recent report based on episome delivery provided a significant technological advance[12]; however, this system may not be sufficient to solve the challenge of limiting the amount and duration of nuclease exposure. Indeed, cells are exposed to

nucleases for several weeks, the necessary time for the appearance of antibiotic-resistant colonies. To date, no studies have investigated a potential deleterious impact of long-term Cas9 expression in microalgae. If that were the case, the transient expression of the CRISPR-Cas9 system described here could decrease undesired mutations compared to long-term expression[29]. A major benefit of RNP delivery is its simplicity and ease of implementation, as it eliminates all subcloning steps, saving considerable time relative to traditional genome-editing approaches.

Another attractive perspective relies on the fact that the *UMPS* and *APT* markers are well conserved within the microalgae phylogenetic tree and among other eukaryotic groups, making it highly likely that the method is extendable to other organisms. This is particularly important for hard-to-transfect species or those for which no biobricks (promoters, terminators) are available.

## Methods

**Culture conditions**. The *Phaeodactylum tricornutum* strain CCMP2561 (NCMA) was grown axenically at 20 °C in vented cap flasks containing silica-free F/2 medium (Sigma G0154) with 40% Sea Salts (Sigma S9883). Incubators were equipped with white neon light tubes providing an illumination of approximately 120 μmol photons m$^{-2}$ s$^{-1}$ and a photoperiod of 12 h light/12 h dark.

**Selection procedures**. For the Nourseothricin (NAT) selection procedure, cells transformed with the NAT selection plasmid were collected two or four days post-transformation and spread on two F/2 agar plates with 300 μg ml$^{-1}$ NAT (Werner Bioagents). After three weeks, colonies were re-streaked on fresh 10-cm 1% agar plates containing 300 μg ml$^{-1}$ NAT. For selection on 5-FOA, NAT-transformants were then transferred onto F/2 agar plates containing 50 μg mL$^{-1}$ uracil and 100 μg mL$^{-1}$ 5-FOA. In RNP experiments, cells were spread two or four days post-transformation onto F/2 agar media supplemented for selection with 50 μg mL$^{-1}$ uracil and 100 μg mL$^{-1}$ 5-FOA, to detect UMPS-mutated colonies, or onto F/2 agar media supplemented with 5 mg L$^{-1}$ adenine and 10 μM of 2-FA, to detect APT-mutated colonies. In the RNP experiments simultaneously targeting the UMPS, APT, and Aureochrome1A genes, cells were selected on F/2 medium supplemented with 10 μM 2-FA, 5 mg L$^{-1}$ adenine, and 50 μg mL$^{-1}$ uracil.

**TALEN nuclease**. The TALEN pairs were designed using TALE-NT (www.tale-nt.cac.cornell.edu)[30,31]. We included two 20-bp recognition sites separated by an 18-bp spacer (TGTCAAAACATAATACCGGAtgatgtgccgattttgttGGATGTCAAGCG CGGCGACA)[30]. All TALEN constructs were assembled through Golden Gate cloning and cloned into TALEN scaffolds[6].

**CRISPR-Cas9 design and constructs**. Guide RNAs (gRNAs) were selected using the CRISPOR tool (http://crispor.tefor.net/)[25], based on the Moreno-Mateos score and the absence of predicable off targets. Corresponding crRNA sequences are listed in Supplementary Table 3. The gRNAs were cloned under control of the U6 promoter in a vector derived from pKSdiaCas9[10]. This vector was co-delivered with the Cas9 encoding plasmid pKSdiaCas9.

**Biolistic transformation of TALEN or Cas9 and gRNA vectors**. Cells ($1.5 \times 10^8$ total) were collected from exponentially growing cultures and spread onto 1% agar plates containing F/2 medium with 20 g L$^{-1}$ sea salt. Transformations were carried out 24 h later using the microparticle bombardment method adapted from Apt et al. [32], with minor modifications. Gold particles (0.6 μm diameter, BioRad) were coated with DNA using 1.25 M CaCl$_2$ and 20 mM spermidine. As a negative control, beads were coated with 3 μg NAT selection plasmid and 3 μg empty vector. For the CRISPR-Cas9 experiment, the DNA mixture contained 3 μg Cas9 expression vector (pKSdiaCas9)[10], 3 μg U6-gRNA encoding plasmid, and 3 μg NAT selection plasmid. As a negative control, beads were coated with a DNA mixture consisting of 3 μg NAT selection plasmid, 3 μg Cas9 expression vector, and 3 μg empty vector. A burst pressure of 1550 psi and a vacuum of 25 Hg were used.

**Preparation of the crRNA–tracrRNA duplex**. Purified crRNAs and tracrRNA were purchased from IDT. Duplexes were prepared following the manufacturer's instructions.

**In vitro CRISPR-Cas9 RNP cleavage assay**. DNA fragments containing the target site were amplified, purified, and eluted with RNase-free water. CRISPR-Cas9 RNP complexes were assembled using crRNA::tracrRNA duplexes and HiFi Cas9 (IDT #1074181). Each RNP complex was combined with the corresponding DNA amplicon and the in vitro cleavage reaction allowed to proceed, following the manufacturer's instructions. Sequential RNase A and proteinase K treatments were performed before gel separation and visualization.

**Biolistic bombardment of Cas9 RNP complexes**. RNP complexes were assembled following the instructions of IDT and delivered to WT *Phaeodactylum* cells by particle bombardment. Briefly, 24 h before the experiment, cells ($1.5 \times 10^8$) were spread onto 1% agar F/2 plates with 20 g L$^{-1}$ sea salt and 50 μg mL$^{-1}$ uracil, if targeting *PtUMPS*. For each shot, the equivalent of 4 or 8 μg Cas9 protein (for multiple targets, the total amount was split equally between the different RNPs) in a total volume of 8 μl Cas9 reaction buffer (NEB, B0386A) was mixed with 10 μl gold nanoparticles (3 mg, 0.6 μm in diameter, Bio-Rad) washed twice with Cas9 reaction buffer. The coated particles were distributed slowly in a circle directly onto the macrocarrier, which was allowed to dry for 2 h before biolistic bombardment. A burst pressure of 1550 psi and a vacuum of 25 Hg were used.

**Genotypic characterization**. Cell lysates served as the PCR templates[6]. Primers listed in Supplementary Table 5 were used to amplify the various loci of interest. The Q5 High Fidelity DNA polymerase (New England BioLabs, USA) was used for general screening purposes. The highly selective HiDi DNA polymerase (myPOLS, Germany) was used for allele-specific PCR, together with primers ending with single nucleotide polymorphisms.

**Deep sequencing**. To quantify the frequency of mutagenic events, locus-specific PCR products were sequenced using NGS technology (S5 Thermofisher). In all conditions, 13,000–72,000 reads per sample were analyzed. The sequencing and the processing of NGS data was done by the GeT-Biopuces platform (INSA Toulouse) using the Ion Torrent Suite software analysis. The percentage of mutagenic event was calculated at position 145 (3nt upstream of the gUMPS1 PAM) and at +/− 5 bases surrounding this position. The frequency of total INDELs (including 1 base INDEL) was calculated, with a background noise level lower than 0.4%. The frequency of INDELs larger than 1 base was also analyzed and in this condition the background noise level was extremely low <0.06%.

**Cell growth experiments**. For the growth experiments reported in Fig. 1e–g, cells were cultivated in F/2 medium for five days and then diluted to 1.10$^5$ cells mL$^{-1}$ in F/2 medium, F/2 medium supplemented with 50 μg mL$^{-1}$ uracil, or F/2 medium supplemented with 50 μg mL$^{-1}$ uracil and 100 μg mL$^{-1}$ 5-FOA. For the growth experiment reported in Fig. 4, a similar procedure was followed, except that cells were diluted into F/2 medium, F/2 medium with 5 mg L$^{-1}$ adenine, or F/2 medium with 5 mg L$^{-1}$ adenine and 10 μM 2-FA after five days.

**Measurement of non-photochemical quenching (NPQ)**. Cell suspensions in mid-exponential phase were adjusted to a chlorophyll a content of 1 μg mL$^{-1}$ and NPQ was measured with an AquaPen-C AP 100 (Photon Systems Instruments, Brno, Czech Republic) using light pulses with an intensity of 2100 μmol photons m$^{-2}$ s$^{-1}$ applied every 20 s to induce maximal fluorescence and 700 μmol photons m$^{-2}$ s$^{-1}$ of actinic light to induce NPQ.

**Protein extractions and immunoblotting**. Cells were grown to mid-exponential phase. Forty milliliters of culture were harvested by centrifugation and lysed in a Savant FastPrep FP120 bead mill (Thermo Scientific, Karlsruhe, Germany)[9]. Approximately twenty-five micrograms of protein were separated in a 10% polyacrylamide gel by SDS-PAGE and blotted onto a nitrocellulose membrane. After blotting, the nitrocellulose membrane (Amersham Protran 0.1 μm NC, GE Healthcare) was cut between 35 and 40 kDa and the top half (40–250 kDa) was used to detect PtAureo1a, whereas the bottom half (0–35 kDa) was used to detect the D1 loading control. Immunoblots to detect PtAureo1a were made using a custom-made antiserum specific against PtAureo1a (Agrisera AB, Vännas, Sweden) at a 1:1000 dilution[9]. The D1-specific antiserum (AS05-084, Agrisera AB) was used according to the manufacturer's instructions, at a 1:20000 dilution. Blots were developed using an Odyssey FC Imaging System (Li-Cor, Bad Homburg, Germany). An uncropped scan of the blot is shown in Supplementary Fig. 15.

## Data availability

Short read sequence data have been deposited at the NCBI Sequence Read Archive (SRA) under the accession code SRP160923 . The authors declare that all data supporting the findings of this study are available within the article and its Supplementary Information files or are available from the corresponding author upon request.

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

## Acknowledgements

This work was funded by a Gordon and Betty Moore Foundation grant (GBMF 4966), a Région Midi-Pyrénées grant (15058490 financial support for Accueil d'Equipes d'Excellence), an ANR JCJC grant (ANR-16-CE05-0006-01), and the 3BCAR Carnot Institute funding. We thank Dr. Fabien Nogué for stimulating scientific discussions, Dr. Peter G. Kroth for providing the PtAureo1a anti-serum, Dr. Jean Pierre Bouly and Dr. Angela Faciatore for discussion on NPQ analyses.

## Author contributions

F.D. conceived the study. M.S., G.D., A.F., D.J. and F.D. designed the experiments. M.S., G.D., A.F., M.T., D.J. and F.D. performed and analyzed the experiments. D.J. and F.D. wrote the manuscript with support from all authors.

## Additional information

**Competing interests:** The authors declare no competing interests.

