## [Peer Review File · Nature Communications]

Reviewers' Comments:

Reviewer #1:

Remarks to the Author:

Review of NCOMMS-18-15857

One-step generation of multiple gene knock-outs in diatoms by DNA-free genome editing

General Comments

This is a potentially useful advance for the algal research community that applies existing DNA free genome editing technologies to demonstrate multiplex genome editing in *Phaeodactylum tricornutum*. Similar studies have been carried out in another model algal species, *Chlamydomonas*. There are two major selling points for the manuscript; (1) the targeting of uracil and adenine biosynthesis genes to create phenotypes compatible with negative selection (5-FOA and 2-FA) that do not rely on use of antibiotics, and (2) multiplexing of guideRNAs with purified Cas9 to create edits in up to three genes at a time. The novelty of this paper is the application to *P. tricornutum* rather than in development of a new genome-editing method.

Overall, the paper is succinct, the data consistent, and fairly straightforward to understand. The authors claim that their methodology is an improvement on existing-plasmid based Cas9 gene editing methods. This may be correct in the sense that the new method does not require cloning, but their methodology does require specialized machinery (for biolistic bombardment) that may not be accessible to all researchers. Furthermore, the authors should show caution in claiming that protein-based gene editing has potential for less toxicity than other methods because of shorter Cas9 half-life, especially considering that 6 sgRNAs are multiplexed in their experiments. A number of studies in mammalian cells have shown that multiplexing sgRNAs is in fact a robust experimental methodology to promote chromosomal translocations or other major rearrangements that cause genome instability and potential cellular toxicity (for example PMID 24759083). Unless the authors provide specific evidence that multiplexing RNP/guideRNAs has less off-target toxicity than other methods, these claims should be tempered. Indeed, the issue of off-target toxicity for Cas9 or other genome-editing nucleases has not been seriously addressed in *P. tricornutum*.

Specific comments:

1. Figure 1. The images of the cultures below the growth curves are hard to visualize, and in some cases don't appear to correspond to the plotted values. On the y-axis, what is Ln cells ml⁻¹? This is not defined in the legend or materials section.
2. Page 5, line 98. For the 36 8A2 and 29 12A1 subclones what proportion were KOs (evidence of editing at both alleles) versus mosaics?
3. Page 7, lines 130-135. The utility of the in vitro cleavage assays is to determine if the sgRNA can direct cleavage to the target gene. From an experimental perspective, this data doesn't really don't add anything as the in vitro assay are not quantitative, and there is no relationship to in vitro efficiency and in vivo editing (which relies on cellular factors, chromatin status, etc..). I would suggest moving these panels in Figures 2 and 4 to the supplement.
4. Page 8, line 150. The statement "Several 5-FOA clones appeared" is vague. How many clones? This is buried in the supplement but is an important point because the utility of the proposed DNA-free editing method relies on the ability to robustly select directly for 5-FOA clones. What is the reproducibility of this experiment? The supplemental data appear to list a single replicate for each sgRNA.
5. Page 8, line 160. The sentence "The kinetics of DSB production and repair in vivo are unknown." seems out of place here, as the following experiments in no way measure kinetics of repair. Please remove or further justify.
6. Page 8, line 169, "...suggesting major rearrangement". Or, a large deletion that extends outside of primer pairs used for amplification (but perhaps this is what the authors mean).

7. Page 9, line 192, replace transformation with bombardment (there is no transformation in this experiment).
8. Data presentation in Figures 3, 4, and 5. I find the pie-charts misleading, and the color or shading schemes are not easy on the eye. What not a stacked barplot of proportion of each event? There is no numerical data provided in the pie chart so it is difficult for the reader to assess magnitude of events relative to each other.
9. For the multiplexed experiments, it is hard to get a sense of reproducibility as the initial colony counts from the bombardment are low, and seem to range in from 3-30 ish colonies. If this technique is to become widespread, then a robust and reproducible protocol that yields significant number of colonies in the hands of non-specialists should be developed. Or, provide experimentalists with some general idea of reproducibility.
10. Page 12. Were the 2/13 colonies that showed editing at all 3 loci screened phenotypically for *ura-*, 5-FOA and 2-FA resistance, and blue colonies?
11. Page 13, line 272. Yes, if this method is reproducible in other labs and with other sgRNAs, then it will be a major advance. Perhaps a note of caution here would be wise?
12. Page 12, line 288. While it is true that plasmid-based systems are harder to regulate than DNA-free delivery systems, I am not aware of evidence showing that continued overexpression of Cas9 is toxic in algae. If the authors are aware of studies showing this then please cite them, otherwise this statement should be toned down.
13. Page 13, line 291. The reference here [ref 30] is for Cas9 stability in mammalian cells, and may not be directly relevant for algae. Did the authors measure half-life of Cas9 in *P. tricornutum*?
14. Page 13, line 293. The authors should acknowledge that multiplexing sgRNAs with Cas9 is a robust method of generating chromosome arrangements in mammalian systems. Surely this would be an issue in algae as well.

Reviewer #2:

Remarks to the Author:

This manuscript described development of multiple genome-editing methods in the diatom *Phaeodactylum tricornutum* by the biolistic delivery of CRISPR Cas /RNP complex. This paper also reported that *PtUMPS* and *PtAPT* genes were represented as endogenous selectable markers for the mutant generation.

However, several major concerns are needed to be clarified for the publication.

Successful genome editing of *Phaeodactylum tricornutum* by using TALEN method (2014) was firstly reported, and CRISPR-CAS from same species is also published (2016), and many CRISPR /CAS related, including CRISPR/ Cpf method, have been introduced in other algae.

I agree that the results of this manuscript nicely described CRISPR/Cas RNP transformation of *P. tricornutum* for genome editing. I am sure this strategic development for combining endogenous makers to generate TM strain of *P. tricornutum* will provide the helpful tool for the strain development or basic research. But the points that authors claimed throughout this manuscript such as simultaneous co-delivery of CRISPR RNP complex to *P. tricornutum* for genome edition seems to be diluted by several previously published papers; Nature Protocol (2018), and Nature Comm (2016) paper described biolistic delivery of CRISPR/CAS RNP for wheat callus cells and maize embryo cells which also have thick cell wall. Besides, the multiplex genome editing results are already seen in another organism (animal) by delivering multiple species of Cas9 RNP (differing only by gRNA target), which was published in Journal of Biotechnology (2015). In this regard, I am concerned about the novelty of this work.

There is the list of additional points to be clarified.

1. It would be essential to know how many CRISPR based in/del mutations of PtAUREO1a obtained via biolistic method without combining selective marker. Readers of this manuscript would like to know the CRISPR RNP derived mutation ratio in diatom as well as off-target effect. For this, authors should perform targeted deep sequencing analysis of bulk samples to calculate in/del ratio.
2. Single or double gene knockout of PtUMPS and PtAPT were assessed well, and the genotypes of PtAureo1a knockout mutants combined with gene knockout of PtUMPS or/and PtAPT, were displayed in Table 4S. The number of the biallelic mutant of PtAureo1a is 4, and monoallelic mutation is 3 out of 13, 2-FA resistant colonies. Are these mutants phenotypically knocked out? Authors did not give any clue of knocked-out phenotypes in manuscript. Although the previous study had already revealed that the TM generated by TALEN method (2017) resulted in phenotypic differences, authors of this paper should show the phenotypic variation of monoallelic and biallelic PtAureo1a mutants chosen from the Table S4 to convince readers that multiplexed based PtAureo1a knockout mutants display the different phenotype of WT.
3. It is curious that 13 of 2-FA resistant clones shown in Table 4S are summated from the several biolistic results or result from the single shot. Need detail explanations
4. From figure 3 to 5, the genotypic characterizations of KO mutants are present in the form of Pie Chart, which did not help to figure out of data. It would be better to present and explain with the results of PCR from KO mutants and together with the number or ratio of each type.
5. In line 247 to 258, it seems that description of the % of an event is not quite matched with the diagram of Fig. 5. Need to be clarified. (For examples, "Forty-seven percent (16/34)" is mismatched with Fig 4f; Fig. 5 appears to be incomplete (In line 251, mentioned Fig 5a. is there b or c missing?).
6. I think Table 4S is the most key data to show in the main text, or it is better to combine Figure 5 with table S4. And it would be better to incorporate the growth curve of all 13 clones compared WT in supplement data.
7. In the title, knock-outs" in diatoms" should be changed to "in *P. tricornutum*, since only this species worked.

Reviewers' comments:

Reviewer #1 (Remarks to the Author):

Review of NCOMMS-18-15857

One-step generation of multiple gene knock-outs in diatoms by DNA-free genome editing

General Comments

This is a potentially useful advance for the algal research community that applies existing DNA free genome editing technologies to demonstrate multiplex genome editing in *Phaeodactylum tricornutum*. Similar studies have been carried out in another model algal species, *Chlamydomonas*. There are two major selling points for the manuscript; (1) the targeting of uracil and adenine biosynthesis genes to create phenotypes compatible with negative selection (5-FOA and 2-FA) that do not rely on use of antibiotics, and (2) multiplexing of guide RNAs with purified Cas9 to create edits in up to three genes at a time. The novelty of this paper is the application to *P. tricornutum* rather than in development of a new genome-editing method.

Author response: We thank the reviewer for his positive comments. It is true that the major achievement of this paper is the ability to perform DNA-free genome editing in the hard-to-transfect *P. tricornutum* cells. We believe that our work is particularly significant because it points out the versatility and power of this new genome engineering strategy for manipulating genetically intractable organisms of biotechnological importance. This strategy has never been reported before, even in organisms where targeted genome editing is well-established. Here, we even go one step further by demonstrating efficient simultaneous multiple gene knock-outs, a first proof in microalgae.

---end response---

Overall, the paper is succinct, the data consistent, and fairly straightforward to understand. The authors claim that their methodology is an improvement on existing-plasmid based Cas9 gene editing methods. This may be correct in the sense that the new method does not require cloning, but their methodology does require specialized machinery (for biolistic bombardment) that may not be accessible to all researchers.

Author response: We agree with the reviewer's comment. The main goal of the manuscript was to establish the proof of concept of DNA-free genome editing in *P. tricornutum* using the recognized and state-of-the-art transformation method, the biolistic particle bombardment. Other transformation methods have been reported for *P. tricornutum* such as electroporation but the results revealed an inconsistency from one laboratory to another. Biolistic is one of the least efficient methods but also the most robust one. We are aware that the apparatus is expensive and not accessible to all. However, the strategy we developed does not require any biobricks and involves a highly conserved counter-selectable marker, making it easily transferable to other algae or other organisms, where other delivery methods may be available. Furthermore, we are working on new RNP delivery systems but achieving robust results can be challenging and time consuming. Therefore, we do not expect results before several months.

---end response---

Furthermore, the authors should show caution in claiming that protein-based gene editing has potential for less toxicity than other methods because of shorter Cas9

half-life, especially considering that 6 sgRNAs are multiplexed in their experiments. A number of studies in mammalian cells have shown that multiplexing sgRNAs is in fact a robust experimental methodology to promote chromosomal translocations or other major rearrangements that cause genome instability and potential cellular toxicity (for example PMID 24759083). Unless the authors provide specific evidence that multiplexing RNP/guideRNAs has less off-target toxicity than other methods, these claims should be tempered. Indeed, the issue of off-target toxicity for Cas9 or other genome-editing nucleases has not been seriously addressed in *P. tricornutum*.

Author response: We agree with the reviewer's comment. We modified the text to temper our claims:

"To date, no studies have investigated a potential deleterious impact of long-term Cas9 expression in microalgae. If that were the case, the transient expression of the CRISPR/Cas9 system described here could decrease undesired mutations compared to long-term expression"

---end response---

Specific

comments:

1. Figure 1. The images of the cultures below the growth curves are hard to visualize, and in some cases don't appear to correspond to the plotted values. On the y-axis, what is Ln cells ml⁻¹?? This is not defined in the legend or materials section.

Author response: We apologize for the lack of information in the text. The Y-axis presented the natural logarithm of cell density but, to improve clarity, we have modified it. In the new version, the Y-axis directly presents cell density (in millions cells per mL), and the axis scale was changed to a logarithmic scale. This information has been added in the figure caption as follows: "Y-axis (**e to g**) presents cell density, expressed as million cells per mL, on a logarithmic scale." We moved the pictures to Supplementary Figure 2. We made a mistake in the first version as the pictures were taken on day 10 and not at day 14, which corresponds well to the graphs.

---end response---

2. Page 5, line 98. For the 36 8A2 and 29 12A1 subclones what proportion were KOs (evidence of editing at both alleles) versus mosaics?

Author response: The reviewer raises a good point. These subclones came from the 8A2 and 12A1 mosaic populations that contain cells with different types of mutagenic events but only a few cells with wild type sequences, as illustrated by the graphs in Figure 1C. Only one allele can be amplified in 35 out of the 36 subclones from population 8A2. This allele always bears INDELS, while the second allele is not detectable and probably corresponds to a large deletion event. The last 8A2 subclone presents both alleles, and both alleles display INDELS. Thus, we can conclude that all the subclones from 8A2 are mutated on both alleles. As expected from the 12A1 graph, the proportion of subclones harboring a mutation on both alleles is high (19/20), suggesting that a bi-allelic *PtUMPS* KO is required to confer 5-FOA resistance. However, the genotypes of the 5-FOA resistant clones from later on RNP experiments (76% monoallelic and 24% bi-allelic) show that the inactivation of one allele is sufficient to confer selection to 5-FOA.

We modified the text as follows:

"As expected, all showed mutagenic events (examples are depicted in Fig. 1d). In 19 of 20 12A1 subclones, both *PtUMPS* alleles showed mutagenic events based on polymorphism patterns present upstream and downstream of the target site (Supplementary Fig. 1). Only one allele was amplified in

35 of 36 8A2 subclones and 1 of 20 12A1 subclones (Fig. 1d), suggesting large insertions or deletions in the other allele^{6,9}, a phenomenon known as loss of heterozygosity (LOH)²⁵. In the detected allele, INDELS were systematically present”.

---end response---

3. Page 7, lines 130-135. The utility of the in vitro cleavage assays is to determine if the sgRNA can direct cleavage to the target gene. From an experimental perspective, this data doesn't really don't add anything as the in vitro assay are not quantitative, and there is no relationship to in vitro efficiency and in vivo editing (which relies on cellular factors, chromatin status, etc..). I would suggest moving these panels in Figures 2 and 4 to the supplement.

Author response: These two panels have been moved to Supplementary Figure 3 and 10. Supplementary figures numbering has been modified accordingly.

---end response---

4. Page 8, line 150. The statement “Several 5-FOA clones appeared” is vague. How many clones? This is buried in the supplement but is an important point because the utility of the proposed DNA-free editing method relies on the ability to robust select directly for 5-FOA clones. What is the reproducibility of this experiment? The supplemental data appear to list a single replicate for each sgRNA.

Author response: For the experiment described on Line 150, twelve 5-FOA-resistant clones were obtained and 10 out of them presented targeted mutagenic events (Supplementary Table1). These values have been added to the text:

“Twelve 5-FOA^R clones appeared. Their genotypic characterization confirmed the presence of mutations adjacent to the PAM sequence in 10 out of 12 cases (Supplementary Table 1), demonstrating that direct selection on 5-FOA can lead to the identification of mutagenic events. The two remaining clones were not mutated at the *PtUMPS* loci, suggesting that they were false positives.”

This particular experiment was based on the genomic integration of the Cas9 and guide RNA expression cassettes to knock-out *PtUMPS*. It tested the possibility to perform direct selection on 5-FOA containing medium. Because the purpose of our paper was more focused on RNP delivery, we performed the genomic integration experiment only once. The obtained positive results prompted us to directly shift to RNP experiments. Three independent UMPS RNP experiments have been performed and several 5-FOA resistant clones were obtained (24 for one experiments and 12 for the second, 10 for the third) and mutagenic events detected at the *PtUMPS* loci. As mentioned in Line 260, the selection of 5-FOA-resistant can fail in some circumstances but these experiments were not taken into account. For the APT RNP, 10 independent experiments have been performed and mutagenic events have been observed for all of them.

5. Page 8, line 160. The sentence “The kinetics of DSB production and repair in vivo are unknown.” seems out of place here, as the following experiments in no way measure kinetics of repair. Please remove or further justify.

Author response: This sentence has been removed.

---end response---

6. Page 8, line 169, "...suggesting major rearrangement". Or, a large deletion that extends outside of primer pairs used for amplification (but perhaps this is what the authors mean).

Author response: We agree with reviewer's comment. For more clarity, we have modified the sentence "We failed to amplify the *PtUMPS* locus in the gUMPS1- and gUMPS3-derived strains, suggesting that a large deletion may have occurred"

---end response---

7. Page 9, line 192, replace transformation with bombardment (there is no transformation in this experiment).

Author response: We have made the change.

---end response---

8. Data presentation in Figures 3, 4, and 5. I find the pie-charts misleading, and the color or shading schemes are not easy on the eye. What not a stacked barplot of proportion of each event? There is no numerical data provided in the pie chart so it is difficult for the reader to assess magnitude of events relative to each other.

Author response: To improve clarity, we modified Figures 3, 4 and 5 and their corresponding legends. We used stacked barplots in Figures 3c and 3d to describe the distribution of the mutagenic events classes. We changed the color schemes in Figures 3, 4 and 5. We added numerical values for the repartition of classes in Figures 4 and 5. Figure legends were modified accordingly.

---end response---

9. For the multiplexed experiments, it is hard to get a sense of reproducibility as the initial colony counts from the bombardment are low, and seem to range in from 3-30 dish colonies. If this technique is to become widespread, then a robust and reproducible protocol that yields significant number of colonies in the hands of non-specialists should be developed. Or, provide experimentalists with some general idea of reproducibility.

Author response: We are aware of the necessity to share a robust protocol for the technique to become widespread. The transformation efficiency of *P. tricornutum* by biolistic is very low for DNA (around 10^{-7}) as well as for protein delivery, explaining the low number of clones obtained. The major point is that we reliably manage to generate a dozen of transgenic strains per shot, an important proportion of them (66-100%) being co-edited. A table summarizing results from the independent experiments that were performed has been added for clarification (Supplementary Table 4), as shown below.

Biolistic experiment number	Targeted genes	Selection conditions	Number of colonies mutated on the marker	Number of colonies mutated on at least one of the other target
#1	UMPS + Aureo1A	5-FOA	17	17/17 (100%)
#2	APT + Aureo1A	2-FA	29	19/29 (66%)
#3	APT + UMPS + Aureo1A	2-FA	13	10/13 (77%)

“The major point is that we have succeeded in generating multiple PtAureo1a knock-out strains in the three independent multiplexed experiments described in Supplementary Table 4.”

---end response---

10. Page 12. Were the 2/13 colonies that showed editing at all 3 loci screened phenotypically for ura-, 5-FOA and 2-FA resistance, and blue colonies?

Author response:

As suggested by both reviewers, we phenotypically characterized a total of 11 transgenic strains coming from multiple independent RNP experiments. Results are summarized in Supplementary Figures 14 to 17. The 2/13 clones that showed editing in all 3 loci were well 2-FA resistant and 5-FOA resistant, as revealed by plating assays (Supplementary Figure 14b). For all the strains, sensitivity or resistance to different media (F2; 5-FOA+ Uracil and 2-FA + Uracil) reflected genotypes well (Supplementary Figure 14a)

The inactivation of Aureo1A does not render cells blue. Instead, it negatively impacts their ability to dissipate excess light energy upon a shift from low to high irradiances (Serif et al., 2017). This can be quantified based on chlorophyll fluorescence measurements that allow calculating the Non-Photochemical Quenching parameter (NPQ). The higher NPQ is, the more efficient cells are at dissipating excess light energy and at performing photoprotection. It was previously shown that biallelic *P. tricornutum PtAureo1a* mutant strains generated with TALE nucleases displayed reduced NPQ values compared to wild-type (Serif et al., 2017). We performed similar NPQ measurements and observed a 50%-73% reduction of NPQ in the biallelic *PtAureo1a* mutants generated through RNP compared to controls (Supplementary Figure 17). The low NPQ phenotype was not observed in the monoallelic *PtAureo1a* mutant strain tested. We quantified the amount of Aureo1A in all the strains by Western Blotting. We did not detect any Aureochrom1A protein in the samples corresponding to the biallelic mutated strains. The Aureochrom1A protein is detected in the sample corresponding to the mono-allelic mutant strain as well as in the samples for positive controls (APTb1, APTb3, UMPS4 and NCMA).

We also recorded cell density as a function of time to establish growth curves (Supplementary Figure 16). We confirm that the inactivation of *PtUMPS* or *PtAPT* has no deleterious effect on cultures fitness. There was no clear effect of the *PtAureo1a* mutation on growth rates either.

A paragraph has been added to the main text to describe these phenotypic observations.

“To go one step further, we phenotypically characterized eleven RNP-derived transformants from several independent experiments that were mutated in *PtUMPS* and/or *PtAPT* and/or *PtAureo1a* (Supplementary Figure 14a). First we tested their resistance to 5-FOA as well as 2-FA (Supplementary Figure 14b). For all these strains, sensitivity or resistance to different media reflected genotypes well. Next, we quantified the amount of PtAureo1a in the strains by Western Blotting (Supplementary Figure 15). Whereas we did not detect any PtAureo1a protein in the samples derived from bi-allelic *PtAureo1a* mutant strains, we clearly detected it in protein samples derived from the monoallelic PtAureo1a strain and the positive controls. We additionally performed growth experiments and observed that the mutations did not have any strong negative impact on culture fitness (Supplementary Figure 16).

Finally, we evaluated the photo-physiological impact of the PtAureo1a mutation. This gene encodes for a blue-light photoreceptor of which knock-out strongly affects cells ability to dissipate excess light energy as heat upon a shift from low to high irradiances⁹. This can be quantified by measuring the Non-Photochemical Quenching (NPQ) of chlorophyll fluorescence⁹. As previously reported biallelic

P. tricornutum *PtAureo1a* mutant strains generated by TALE nucleases display reduced NPQ capacities compared to wild-type⁹. We performed similar NPQ capacity measurements. Whereas no NPQ phenotype was observed in the monoallelic *PtAureo1a* mutant strain tested here and in the PtAPT and PtUMPS knock-out strains, we measured a 50%-73% reduction of NPQ in all RNP-derived biallelic *PtAureo1a* mutants tested (Supplementary Figure 17). Therefore, we confirmed the loss of function of the PtAureo1A knock-out strains generated which reveals the power of our approach to study gene function.”

The Methods section has also been updated:

“Measurement of non-photochemical quenching (NPQ). Cell suspensions in mid-exponential phase were adjusted to a chlorophyll a content of $1\mu\text{g mL}^{-1}$ as previously reported⁹ and NPQ was measured with an AquaPen-C AP 100 (Photon Systems Instruments, Brno, Czech Republic) using light pulses with an intensity of $2100\mu\text{mol photons m}^{-2}\text{ s}^{-1}$ applied every 20s to induce maximal fluorescence and $700\mu\text{mol photons m}^{-2}\text{ s}^{-1}$ of actinic light to induce NPQ.

Protein extractions and immunoblotting. Cell culturing, protein extractions and immunoblotting were performed as previously described⁹. Immunoblots using a custom-made antiserum specific against Aureochrome 1a (Agrisera AB, Vännas, Sweden) were performed, whereas the D1-specific antiserum (AS05-084, Agrisera AB) was used according to the manufacturer’s instructions.”

Altogether, these results reinforce the power of our approach to study gene function

---end response---

11. Page 13, line 272. Yes, if this method is reproducible in other labs and with other sgRNAs, then it will be a major advance. Perhaps a note of caution here would be wise?

Author response: This sentence was supported by our enthusiasm for this new approach and the potential for elucidating complex mechanisms. We have modified the sentence as follows: “This is a major advance for algae genome engineering and should open the way to the study of putative redundant functions of multiple gene family members. For example, it could be suitable to investigate the role of specific light-harvesting complexes of the LHCX family during exposure to excess light²⁹ or the contribution of single aureochromes to photoacclimation¹¹.”

---end response---

12. Page 12, line 288. While it is true that plasmid-based systems are harder to regulate than DNA-free delivery systems, I am not aware of evidence showing that continued overexpression of Cas9 is toxic in algae. If the authors are aware of studies showing this then please cite them, otherwise this statement should be toned down.

Author response: We agree with the reviewer’s comment.

We have modified this sentence as follow: “To date, no studies have investigated a potential deleterious impact of long-term Cas9 expression in microalgae. If that were the case, the transient expression of the CRISPR/Cas9 system described here could decrease undesired mutations compared to long-term expression.”

---end response---

13. Page 13, line 291. The reference here [ref 30] is for Cas9 stability in mammalian cells, and may not be directly relevant for algae. Did the authors measure half-life of Cas9 in *P. tricornutum*?

Author response: In mammals, exogenously added CAS9 protein half-life varies from a few hours to several days³⁰. This parameter is not easily measurable in *P. tricornutum* cells because of the low transformation efficiencies achieved (less than 10^{-6}). However, we can assume that the CAS9 protein will be degraded within the few days following RNP delivery.

---end response---

14. Page 13, line 293. The authors should acknowledge that multiplexing sgRNAs with Cas9 is a robust method of generating chromosome arrangements in mammalian systems. Surely this would be an issue in algae as well.

Author response: It is true that multiplex gRNA can induce chromosomal rearrangements. However, if such translocations occurred, we should not be able to amplify the locus by PCR. We are aware of this potential drawback and we will solve the issue by screening several clones. To be consistent, the phenotype of 5 biallelic knock-out PtAureo1a strains have been investigated and all of them display a similar NPQ reduction.

---end response---

Reviewer #2 (Remarks to the Author):

This manuscript described development of multiple genome-editing methods in the diatom *Phaeodactylum tricornutum* by the biolistic delivery of CRISPR Cas /RNP complex. This paper also reported that PtUMPS and PtAPT genes were represented as endogenous selectable markers for the mutant generation.

However, several major concerns are needed to be clarified for the publication.

Successful genome editing of *Phaeodactylum tricornutum* by using TALEN method (2014) was firstly reported, and CRISPR-CAS from same species is also published (2016), and many CRISPR /CAS related, including CRISPR/ Cpf method, have been introduced in other algae.

I agree that the results of this manuscript nicely described CRISPR/Cas RNP transformation of *P. tricornutum* for genome editing. I am sure this strategic development for combining endogenous makers to generate TM strain of *P. tricornutum* will provide the helpful tool for the strain development or basic research. But the points that authors claimed throughout this manuscript such as simultaneous co-delivery of CRISPR RNP complex to *P. tricornutum* for genome edition seems to be diluted by several previously published papers; Nature Protocol (2018), and Nature Comm (2016) paper described biolistic delivery of CRISPR/CAS RNP for wheat callus cells and maize embryo cells which also have thick cell wall.

Besides, the multiplex genome editing results are already seen in another organism (animal) by delivering multiple species of Cas9 RNP (differing only by gRNA target), which was published in Journal of Biotechnology (2015). In this regard, I am concerned about the novelty of this work.

Author response: Several studies have recently been published on RNP delivery, this in a variety of organisms. What we are emphasizing here is the selection strategy to enrich for RNP-transformed, co-edited cells when transformation efficiency is extremely low. We perhaps did not sufficiently highlight the difficulty or impossibility to deliver DNA or proteins into *P. tricornutum* cells without using an antibiotic marker. We therefore added a few sentences in this sense.

“We did not expect a RNP transformation efficiency higher than 10^{-6} in the absence of selection, as in the case of DNA delivery it is necessary to co-transform 100 million cells with a plasmid encoding an antibiotic resistance to obtain 30 antibiotic-resistant clones^{6,9,10}.”

Until now, all DNA transformation experiments performed in *P. tricornutum* have necessitated the co-delivery of a plasmid encoding an antibiotic resistance marker. Using antibiotic selection, bombardment transformation efficiencies range from 10^{-7} - 10^{-6} cells μg^{-1} DNA, meaning that for 150 million bombarded cells between 15 to 150 antibiotic-resistant colonies appear. There is no reason to believe that protein delivery would be much more effective than DNA delivery. So, RNP delivery in the presence of a vector encoding an antibiotic resistance marker should at best have the same efficiency, even if no one has yet published it in *P. tricornutum*. Moreover, not all the strains displaying antibiotic resistance will have integrated the nuclease. In this paper, we have successfully tackled several challenges:

1. Transform the alga *P. tricornutum* for the first time without using an antibiotic vector and obtain transgenic strains.
2. Identify new counter-selectable marker genes whose inactivation confer positive selection

3. Use these new markers to establish the proof of concept of RNP delivery
4. Generate simultaneously in one-step triple knock-out strains, whereas multiplexing – even through the nuclear integration of an expression cassette for the CRISPR/Cas9 system – has never been achieved in this species, diatoms or microalgae in general to our knowledge.

As suggested by both reviewers, a phenotypic characterization of the *PtAureo1a* mutants has been performed and added to the manuscript. We could confirm the observations that were previously made on PtAureo1a KOs generated with TALE nucleases (Serif et al., 2017) in terms of impacts on photo-physiology. These results reinforce the power of our approach to study gene function.

The originality and versatility of our approach, combined with the possibility to extend it to other organisms with low transformation efficiencies, prompted us to submit this manuscript to the world-renowned Nature communications journal to reach a broader audience.

---end response---

There is the list of additional points to be clarified.

1. It would be essential to know how many CRISPR based in/del mutations of PtAUREO1a obtained via biolistic method without combining selective maker. Readers of this manuscript would like to know the CRISPR RNP derived mutation ratio in diatom as well as off-target effect. For this, authors should perform targeted deep sequencing analysis of bulk samples to calculate in/del ratio.

Author response: As explained hereinabove, we did not expect RNP transformation efficiency to be higher than approximately 10^{-7} to 10^{-6} . In this scenario, without selection it would be necessary to screen more than one million cell-derived colonies before identifying a mutant strain. With our RNP experiments and selection strategy, we generally get 10-20 colonies per 150 million cells transformed, which corresponds to an efficiency of around 10^{-7} . To demonstrate the difficulty of producing transgenic strains in *P. tricornutum* without an antibiotic marker or any other mean of selection, we performed a targeted deep sequencing experiment as asked by the reviewer. We bombarded WT *P. tricornutum* cells with a RNP targeting *PtUMPS* (first experiment, increasing amounts of RNP gUMPS1: 2µg, 4µg or 8µg) or with a Cas9 only. Cells were collected four days post-transformation and the presence of mutations has been analyzed using a locus-specific PCR followed by Deep sequencing. Corresponding results have been included in Supplementary table 7.

A paragraph describing this experiment has been added to the main text:

“We first evaluated the frequency of targeted mutagenesis induced by the Cas9-gUMPS1 RNP complex in the absence of selection. To achieve that, *P. tricornutum* cells were bombarded with a dose response of 0, 2, 4 and 8 µg of Cas9-gUMPS1 complex. We did not expect a RNP transformation efficiency higher than 10^{-6} in the absence of selection, as in the case of DNA delivery it is necessary to co-transform 100 million cells with a plasmid encoding an antibiotic resistance to obtain 30 antibiotic-resistant clones^{6,9,10}. To test this hypothesis, we collected the cells growing in the absence of selection at four days post-bombardment and quantified the mutagenesis at the *PtUMPS* locus using locus-specific PCR followed by deep sequencing (Supplementary Figure 7). As positive controls, different amounts of a monoallelic mutant strain carrying a 1nt deletion at the gUMPS1 target site were mixed with WT cells to get cell-to-cell ratios of 100%, 10%, 1% and 0, 1% and 0%. Whereas mutagenic

events were detected in the positive controls at the expected frequencies, no induced-mutagenic event was detected in the samples corresponding to cells bombarded with Cas9-gUMPS1.”

The methods section has been updated:

“Deep sequencing. To quantify the frequency of mutagenic events, locus-specific PCR products were sequenced using NGS technology (S5 ThermoFisher). In all conditions, 13,000–72,000 reads per sample were analyzed. The sequencing and the processing of NGS data was done by the Get-Biopuces platform (INSA Toulouse) using the Ion Torrent Suite software analysis. The percentage of mutagenic event was calculated at position 145 (3nt upstream of the gUMPS1 PAM) and at +/- 5bases surrounding this position. The frequency of total INDELS (including 1 base INDEL) was calculated, with a background noise level lower than 0,4%. The frequency of INDELS larger than 1 base was also analyzed and in this condition the background noise level was extremely low < 0,06%.”

Regarding the interrogation about off-targets. We would like to emphasize that we specifically picked gRNAs with no predicted off-targets within *P. tricornutum* genome, using available online tools. Classically, off-target effects quantification has been done by selecting gRNAs with predicted off targets and performing targeted deep sequencing of these loci. Collecting the corresponding results would require to design new gRNAs, validate them in vivo and to rerun deep sequencing experiments, which would take several months and seems beyond the scope of the present manuscript. We are part of a consortium where another laboratory is addressing this issue. The results will not be published before several months. We hope the reviewer will understand that we cannot compete with this laboratory.

---end response---

2. Single or double gene knockout of PtUMPS and PtAPT were assessed well, and the genotypes of PtAureo1a knockout mutants combined with gene knockout of PtUMPS or/and PtAPT, were displayed in Table 4S. The number of the biallelic mutant of PtAureo1a is 4, and monoallelic mutation is 3 out of 13, 2-FA resistant colonies. Are these mutants phenotypically knocked out? Authors did not give any clue of knocked-out phenotypes in manuscript. Although the previous study had already revealed that the TM generated by TALEN method (2017) resulted in phenotypic differences, authors of this paper should show the phenotypic variation of monoallelic and biallelic PtAureo1a mutants chosen from the Table S4 to convince readers that multiplexed based PtAureo1a knockout mutants display the different phenotype of WT.

Author response: As suggested by both reviewers, a phenotypic characterization of several mutants has been carried out as shown in Supplementary Figures 14 to 17. Our main conclusion is that the tested RNP-derived *PtAureo1a* biallelic KOs do display a reduced photoprotective capacity (Non-Photochemical Quenching parameter), as previously reported by Serif and coworkers. Only one monoallelic mutant has been analysed and it presents a NPQ activity comparable to controls. Altogether, these results reinforce the power of our approach to study gene function

A paragraph has been added to the main text to describe these phenotypic observations.

“To go one step further, we phenotypically characterized eleven RNP-derived transformants from several independent experiments that were mutated in *PtUMPS* and/or *PtAPT* and/or *PtAureo1a* (Supplementary Figure 14a). First we tested their resistance to 5-FOA as well as 2-FA (Supplementary Figure 14b). For all these strains, sensitivity or resistance to different media reflected genotypes well. Next, we quantified the amount of PtAureo1a in the strains by Western Blotting (Supplementary Figure 15). Whereas we did not detect any PtAureo1a protein in the samples derived from bi-allelic

PtAureo1a mutant strains, we clearly detected it in protein samples derived from the monoallelic *PtAureo1a* strain and the positive controls. We additionally performed growth experiments and observed that the mutations did not have any strong negative impact on culture fitness (Supplementary Figure 16).

Finally, we evaluated the photo-physiological impact of the *PtAureo1a* mutation. This gene encodes for a blue-light photoreceptor of which knock-out strongly affects cells ability to dissipate excess light energy as heat upon a shift from low to high irradiances⁹. This can be quantified by measuring the Non-Photochemical Quenching (NPQ) of chlorophyll fluorescence⁹. As previously reported biallelic *P. tricornutum PtAureo1a* mutant strains generated by TALE nucleases display reduced NPQ capacities compared to wild-type⁹. We performed similar NPQ capacity measurements. Whereas no NPQ phenotype was observed in the monoallelic *PtAureo1a* mutant strain tested here and in the *PtAPT* and *PtUMPS* knock-out strains, we measured a 50%-73% reduction of NPQ in all RNP-derived biallelic *PtAureo1a* mutants tested (Supplementary Figure 17). Therefore, we confirmed the loss of function of the *PtAureo1A* knock-out strains generated which reveals the power of our approach to study gene function.”

The Methods section has also been updated:

“Measurement of non-photochemical quenching (NPQ). Cell suspensions in mid-exponential phase were adjusted to a chlorophyll a content of $1\mu\text{g mL}^{-1}$ as previously reported⁹ and NPQ was measured with an AquaPen-C AP 100 (Photon Systems Instruments, Brno, Czech Republic) using light pulses with an intensity of $2100\mu\text{mol photons m}^{-2}\text{ s}^{-1}$ applied every 20s to induce maximal fluorescence and $700\mu\text{mol photons m}^{-2}\text{ s}^{-1}$ of actinic light to induce NPQ.

Protein extractions and immunoblotting. Cell culturing, protein extractions and immunoblotting were performed as previously described⁹. Immunoblots using a custom-made antiserum specific against Aureochrome1a (Agrisera AB, Vännas, Sweden) were performed, whereas the D1-specific antiserum (AS05-084, Agrisera AB) was used according to the manufacturer’s instructions.”

---end response---

3. It is curious that 13 of 2-FA resistant clones shown in Table 4S are summated from the several biolistic results or result from the single shot. Need detail explanations

Author response: These 13 2-FA resistant clones result from one biolistic experiment with two shoots. We have carried out several independent experiments leading to the production of *PtAureo1a* knock-out strains as described in the newly added Supplementary Table 4. The *PtAureo1A* linked phenotype has been investigated in strains from two independent experiments.

---end response---

5. From figure 3 to 5, the genotypic characterizations of KO mutants are present in the form of Pie Chart, which did not help to figure out of data. It would be better to present and explain with the results of PCR from KO mutants and together with the number or ratio of each type.

Author response:

As suggested by the reviewers and to improve clarity, we modified Figures 3, 4 and 5 and their corresponding legends. We used stacked barplots in Figures 3c and 3d to describe the distribution of the mutagenic events classes. We changed the color schemes in Figures 3, 4 and 5. We added numerical values for the repartition of classes in Figures 4 and 5. Figure legends were modified accordingly.

---end response---

6. In line 247 to 258, it seems that description of the % of an event is not quite matched with the diagram of Fig. 5. Need to be clarified. (For examples, “Forty-seven percent (16/34)” is mismatched with Fig 4f; Fig. 5 appears to be incomplete (In line 251, mentioned Fig 5a. is there b or c missing?).

Author response: We thank the reviewer for indicating this mismatch. This is an error that we have corrected with the adequate values. Indeed, 34 2-FA resistant clones were obtained but only 29 were genetically characterized.

The text has been modified as follows: “These results were confirmed by sequencing. We also investigated the presence of mutagenic events at the *PtAureo1a* locus. Sixty-five percent (19/29) of the colonies harboring *PtAPT* mutations also showed mutagenic events at the *PtAureo1a* locus and 52% (10/19) of them exhibited a deletion of the fragment between the two target sites. Interestingly, 52% (10/19) of the *PtAureo1a* mutants were mutated on both alleles, leading to the generation of *PtAureo1a* KO strains. (Fig. 4f).”

---end response---

6. I think Table 4S is the most key data to show in the main text, or it is better to combine Figure 5 with table S4. And it would be better to **incorporate the growth curve of all 13 clones compared WT in supplement data.**

Author response: We have fused the information present in the Table 4S and the Figure 5. We performed growth experiments on eleven transgenic strains, including 5 biallelic Aureochrom1A mutant strains, 1 monoallelic mutant and 4 controls cell lines: 1 corresponding to NCMA parental strain, 1 from UMPS RNP experiment, 2 clones from APT RNP experiment. No clear phenotype was observed, some strains appear to have slowed growth but this result is not observed for all Aureochrom1A mutated strains.

We added a sentence in the main text describing these results:

“We additionally performed growth experiments and observed that the mutations did not have any strong negative impact on culture fitness (Supplementary Figure 16).”

---end response---

7. In the title, knock-outs” in diatoms” should be changed to “in *P. tricornutum*, since only this species worked.

Author response: We changed the title as follows: “One-step generation of multiple gene knock-outs in the diatom *Phaeodactylum tricornutum* by DNA-free genome editing”

---end response---

Reviewers' comments 20th August 2018:

Reviewer #1 (Remarks to the Author):

This is a revised version of a manuscript that I previously reviewed. I had a number of comments regarding data presentation in the initial submission that the authors have largely addressed in the revised version.

Author response: We thank the reviewer for his positive comments.

---end response---

However, the choice of color and pattern scheme for some of the bar charts and plots makes it very difficult to distinguish, and impossible if viewing in black and white instead of color. Please consider revising these.

Author response: We used a color code so that readers can find their way around more easily (PtAPT in orange, PtUMPS in blue and PtAureo1a in green). However, we do understand that data must be analyzable after printing in black and white. We therefore modified some of the graphs in Figures 3 and 4 to go in this direction.

---end response---

The authors also addressed concerns regarding reproducibility and robustness of the methodology.

Author response: We thank the reviewer for his positive comments.

---end response---

Replication by other research groups will address whether this methodology “..is a major advance for algae genome engineering..” as the efficiency of generating initial 5-FOA resistant clones is still very low and dependent on specialized biolistic bombardment equipment.

Author response: We modified the sentence as follows: “This is a valuable research tool for algae genome engineering that should open the way to the study of putative redundant functions of multiple gene family members”.

---end response---

Reviewer #2 (Remarks to the Author):

The revised manuscript is better than the previous version was. The authors have revised the manuscript and addressed the reviewers' concerns. Also, they made efforts to add essential experiments and show the data.

I do not doubt that authors showed achievement of technical advances to realize the RNP based Transformation of *P. tricornutum* via endogenous selection marker with multiplexed knockout strategy. So appropriate phenotypic characterizations of multiplexed knockout mutants should be conferred, which will support the advantage of newly employed technique in this study.

Author response: We thank the reviewer for his positive comments.

---end response---

I have some comments related to the revised manuscript.

In supplementary data, the growth curve of mutants seems to be lousy, and authors describe that “Mutations at the *PtAureo1a* loci do not have any impact on growth” on the legend of fig.S 16. But mono and bi-allelic mutation of at *PtAureo1a* loci are likely to affect the growth behavior in both growth rate and maximum cell density, shown in fig.S 16 B C D.

Independent duplicated experiments gave almost identical results of growth rate, but the average values of *PtAureo1a* mutants are lower (~20%) than average values of WT or parental type (Fig s16). Without standard error or SD, it is also hard to criticize. This needs proper explanation or should redo this growth experiment.

Author response: We agree with the reviewer’s comment. It is true that the monoallelic and biallelic *PtAureo1a* mutants displayed approximately 20% lower growth rates compared to the parental and single mutant controls. However, we note that strain AUA14 – which does not harbor any targeted mutagenesis at the *PtAureo1a* loci – grows similarly to the *PtAureo1a* mutants. We modified the main text as follows: “We additionally performed growth experiments and observed that the cells mutated in *PtAureo1A* had slightly lower (20% maximum) growth rates compared to the parental strain or the *PtUMPS* and *PtAPT* single mutants. (Supplementary Figure 16)”.

---end response---

I assumed that the defect of light signaling could disturb both photosynthesis and also NPQ which will affect general growth behaviors. In Serif et al., (2017), knockout mutants of *PtAureo1a* also showed significant phenotypic changes regarding photosynthesis and cell size as well as NPQ. Please explain the reason for the NPQ of AUA10 and UA8 were not determined in Fig S 17. Since the authors chose the *PtAureo1a* as the target gene to be edited, I would like to suggest that phenotypic results from this multiplexed mutants should be in the main data as figure or table.

Author response: Whereas in most articles only one or two mutants are phenotypically characterized (Serif et al., 2017; Mann et al., 2018), here we characterized six biallelic mutant strains from two independent experiments, one monoallelic mutant and four controls, which required an important amount of work.

We observed very little differences in-between the six biallelic mutant strains, which all displayed NPQ values reduced by 50 to 73% compared to the controls. Given the high reproducibility between

strains, we considered it was not necessary to analyze the two UA8 and AUA10 clones that have close genotypes to the UA17 and AUA7 clones, respectively.

We followed the reviewer's suggestion by moving the NPQ data to the newly added Figure 6 within main text.

---end response---